# A Unique Bilateral Variation of the Extensor Carpi Radialis Longus: A Case Report

**DOI:** 10.3390/jfmk9030109

**Published:** 2024-06-25

**Authors:** Maria Amelia Coello, Lokesh A. Coomar, Meadow Campbell

**Affiliations:** Center for Anatomical Science and Education, Department of Surgery, Saint Louis University School of Medicine, Saint Louis, MO 63104, USA; amelia.coello@slu.edu (M.A.C.); lokesh.coomar@slu.edu (L.A.C.)

**Keywords:** extensor carpi radialis longus, extensor carpi radialis brevis, forearm, tendon, anatomical variation, forearm extensors

## Abstract

A novel combination of variations involving the extensor carpi radialis (ECR) muscle group was observed bilaterally in a 75-year-old female cadaver during routine dissection. An accessory tendon was observed arising from the extensor carpi radialis longus (ECRL) and traveling with the primary tendon through the second compartment of the extensor retinaculum. While the primary tendon inserted on the base of the second metacarpal, as is typical of ECRL, the accessory tendon inserted on the base of the third metacarpal. This insertion is typical of the extensor carpi radialis brevis (ECRB) muscle. Additionally, bilateral agenesis of the ECRB was reported. Thirty-two additional forearms were assessed for similar variations, with none being observed. This combination of variations adds to the literature regarding the ECR muscle group, while also being of interest to clinicians, specifically regarding tendon reconstructive procedures as well as accessing the distal radial artery via the anatomical snuffbox.

## 1. Introduction

The extensor carpi radialis longus (ECRL) and extensor carpi radialis brevis (ECRB) are two muscles that belong to the superficial layer of the posterior compartment of the forearm, also known as the extensor compartment [1,2]. The ECRL typically originates from the lateral supracondylar ridge of the humerus, while the ECRB will originate from the common extensor tendon attached to the lateral epicondyle of the humerus. Both muscles run along the length of the forearm and pass through the second compartment of the extensor retinaculum. The ECRL inserts on the dorsal aspect of the base of the second metacarpal, while the ECRB inserts on the dorsal base of the third metacarpal [1,3]. Both muscles function in unison to extend the wrist and abduct it radially (Figure 1).

While rare, variations in the origins of the radial extensor muscles have been reported. In some cases, the ECRB was observed to have originated from the fascia and tendon of the extensor digitorum (ED), rather than the lateral epicondyle of the humerus [4,5]. There have also been reports in which accessory radial extensor muscles have been observed, with only some mirroring the classical path of the ECR muscles to their insertions [6]. Specifically, three types of variations in the ECR muscles have been described: the extensor carpi radialis intermedius, extensor carpi radialis accessorius, and extensor carpi radialis tertius [7,8,9].

The extensor carpi radialis intermedius originates between the ECRB and ECRL. It will pass through either the second compartment of the extensor retinaculum with the ECR muscles or inhabit a separate compartment, before inserting onto the second or third metacarpal [7,10]. The extensor carpi radialis accessorius can arise from either the fascia superficial to the radial extensors or as a muscular slip from the belly of the ECRL and passes through the second compartment of the extensor retinaculum. The insertion of the extensor carpi radialis accessorius is variable, as the tendon will insert directly onto the abductor pollicis brevis (APB) muscle, or it will bifurcate and insert on both the APB and the first metacarpal [7,8,11,12]. Lastly, the extensor carpi radialis tertius originates between the ECRL and extensor digitorum at the common extensor origin. The tendon of the extensor carpi radialis tertius bifurcates deep to the abductor pollicis longus (APL) and travels through the second compartment of the extensor retinaculum, before inserting on the base of the second and third metacarpals [7,10].

Apart from these three well-documented muscular variations, others have been reported in which additional muscle heads and tendons originate directly from either the ECRL or ECRB [13]. In one case, the ECRL exhibited three muscular heads (lateral, intermediate, and medial) unilaterally, with the medial head joining ECRB [14]. Additionally, a bilateral variation in the ECRL was reported, in which an accessory muscle belly arose from the ECRL before inserting on the first metacarpal [7]. Other cases have described additional heads of muscle originating from the ECRB and inserting at various points, such as the dorsal expansion hood of the second digit and the first metacarpal, with the latter providing some abduction of the thumb [15]. One case reported an additional tendon of ECRB passing superficial to the extensor retinaculum before reaching its insertion [12]. Finally, there have been reports of interconnections between the tendons of ECRL and ECRB [16].

This report describes a unique bilateral variation in the ECR muscle group, involving agenesis of the ECRB while the ECRL tendon bifurcates, passes through the second compartment of the extensor retinaculum, and inserts on the bases of the second and third metacarpals. This bilateral variation presents an opportunity to evaluate the extensor carpi radialis musculature and appreciate its orientation and structure.

## 2. Case Presentation

A novel bilateral variation in the ECR muscle group was observed in the posterior forearms of a 75-year-old female cadaver. Thirty-two additional forearms were assessed for similar variations, with none being observed. All bodies were received through the Saint Louis University (SLU) Gift of Body Program of the Center for Anatomical Science and Education (CASE), with signed informed consent from the donors. The donor was preserved using a solution of isopropyl alcohol, glycol, phenol, and formaldehyde. The CASE gift body program abides by all rules set forth by the Uniform Anatomical Gift Act (UAGA).

All dissection was performed per Grant’s Dissector [17]. Upon routine dissection, a singular ECRL muscle belly was revealed following removal of the skin and superficial fascia within the left posterior forearm (Figure 2). The ECRL arose from its classical origin at the lateral supracondylar ridge of the humerus. Approximately halfway along the forearm, the ECRL muscle belly gave rise to two tendons, which were immediately adjacent to one another. The APL and EPB muscles crossed over the tendons, as they would normally. Both tendons completed their course to the base of the second and third metacarpals via the second compartment of the extensor retinaculum. A tendon of the typical size and orientation of ECRL inserted onto the base of the second metacarpal. However, the smaller tendon deviated obliquely to insert on the base of the third metacarpal. Although this accessory tendon inserted where a typical ECRB would, the tendon lacked the size that would be observed in an ECRB tendon.

Upon completion of the dissection of the left posterior forearm, the right forearm was dissected following the same procedures. Dissection of the right posterior forearm also revealed a single ECRL muscle belly (Figure 3). Two tendons were observed arising from the ECRL muscle and inserted on the second and third metacarpal in the same manner as was observed in the left forearm.

## 3. Discussion

Accessory tendons and muscle heads within the ECRL [7,14], as well as ECRB agenesis [10], have been previously reported. However, this case report is unique in that it describes an accessory tendon arising directly from ECRL and inserting on the dorsal aspect of the base of the third metacarpal, along with ECRB agenesis bilaterally. Despite this, the functions of radial deviation and wrist extension will still persist.

Regardless, the pathways of the tendons of ECRB and ECRL are of clinical importance as they are associated with the anatomical snuffbox [18,19]. The anatomical snuffbox is bounded medially by the EPL tendon and laterally by the APL and EPB tendons. Its contents mainly include the radial artery, as well as superficial branches of the radial nerve and the cephalic vein [1,18]. The radial artery typically courses within the snuffbox itself, passing over the ECRL and ECRB tendons and under the APL, EPB, and EPL tendons before continuing into the hand [20]. An understanding of the variation to either the ECRL or ECRB tendons is crucial as the snuffbox is often used as an anatomical landmark for vascular access via the distal radial artery [21].

The tendons of the radial extensor muscles may also be involved in tendon transfer and reconstructive procedures. Tendon transfer is a common surgical procedure performed to restore extension of the wrist and hand following radial nerve injury. This entails transferring nerve bundles from pronator teres muscle (PT) to the ECRL or ECRB [22]. Additionally, the ECRL tendon has been used in scapholunate ligament reconstructive surgery, in which a strip of the ECRL tendon is used to control the scaphoid flexion and pronation due to scapholunate instability [23,24]. Raising awareness of possible variations in the radial extensor muscles will aid surgeons in successful performance of these procedures.

## 4. Conclusions

A unique case was observed, in which a 75-year-old cadaver exhibited a bilateral series of variations, including an accessory tendon arising from the ECRL as well as ECRB agenesis. The primary tendons inserted on the bases of the second metacarpals, while both accessory tendons inserted on the bases of the third metacarpals. This case report adds to the literature regarding variations within the ECRL and ECRB muscles, while also being of interest to clinicians, specifically regarding tendon transfers and reconstructive procedures of the distal posterior forearm and wrist.

## Figures and Tables

**Figure 1 jfmk-09-00109-f001:**
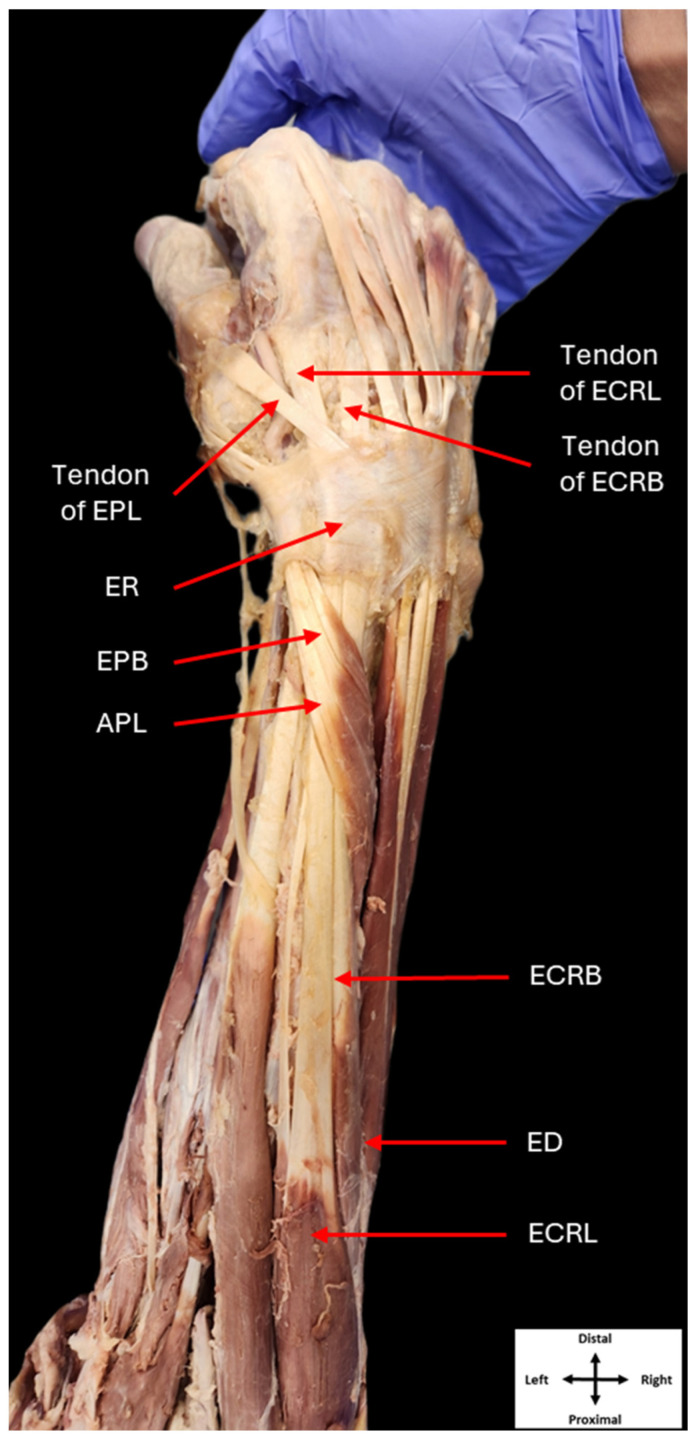
Classical anatomy of the right posterior forearm. Dissection of the posterior compartment of the right forearm showing the typical anatomy of its musculature (ER: extensor retinaculum, ECRL: extensor carpi radialis longus, ECRB: extensor carpi radialis brevis, EPL: extensor pollicis longus, EPB: extensor pollicis brevis, APL: abductor pollicis longus, ED: extensor digitorum).

**Figure 2 jfmk-09-00109-f002:**
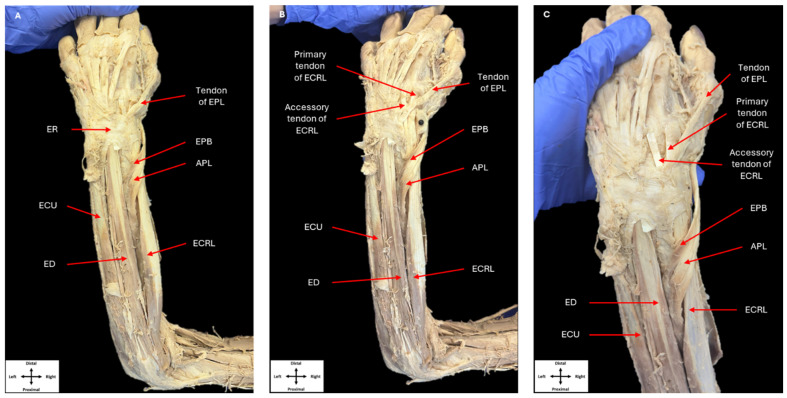
Dissection of the left posterior forearm. Dissection of the posterior compartments of the left forearm showing the muscles in this compartment, as well as their tendons passing beneath the extensor retinaculum (**A**). The extensor retinaculum was removed to show the tendons originating from a single extensor carpi radialis longus muscle belly (**B**,**C**) (ER: extensor retinaculum, ECRL: extensor carpi radialis longus, EPL: extensor pollicis longus, EPB: extensor pollicis brevis, APL: abductor pollicis longus, ECU: extensor carpi ulnaris, ED: extensor digitorum).

**Figure 3 jfmk-09-00109-f003:**
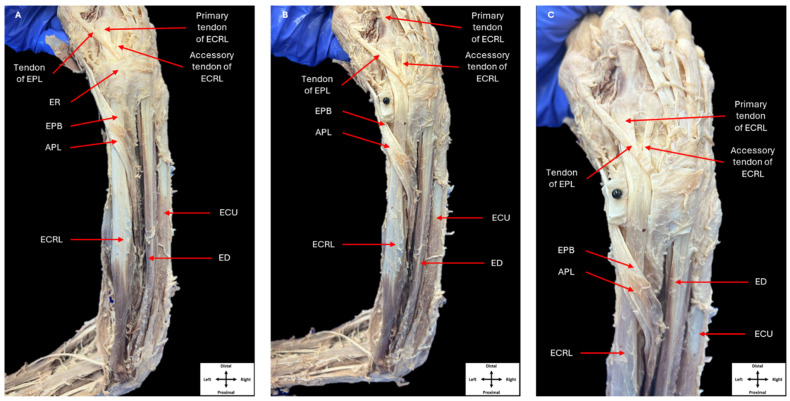
Dissection of the right posterior forearm. Dissection of the posterior compartments of the right forearm showing the muscles in this compartment, as well as their tendons passing beneath the extensor retinaculum (**A**). Removal of the extensor retinaculum allowing the tendons originating from a single extensor carpi radialis longus muscle belly to be observed (**B**,**C**) (ER: extensor retinaculum, ECRL: extensor carpi radialis longus, EPL: extensor pollicis longus, EPB: extensor pollicis brevis, APL: abductor pollicis longus, ECU: extensor carpi ulnaris, ED: extensor digitorum).

## Data Availability

Data are contained within the article.

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
