# Peer review of "A Unique Bilateral Variation of the Extensor Carpi Radialis Longus: A Case Report"

_jfmk, 2024, doi:10.3390/jfmk9030109_

Round 1

Reviewer 1 Report

Comments and Suggestions for Authors

After reading this undoubtedly interesting manuscript I felt exhausted: number of abbreviations used makes reading very boring and difficult to follow.

Second: I have checked several textbooks and there are some concerns about limitation and contents of the anatomical snuffbox: To me it is as follows:

As the snuffbox is triangularly shaped, it has three borders, a floor, and a roof:

  • Ulnar (medial) border: Tendon of the extensor pollicis longus.
  • Radial (lateral) border: Tendons of the extensor pollicis brevis and abductor pollicis longus.
  • Proximal border: Styloid process of the radius.
  • Floor: Carpal bones; scaphoid and trapezium.
  • Roof: Skin.

while Authors find tendons of ECRL and ECRB within it.

Reviewer 2 Report

Comments and Suggestions for Authors

Dear authors,

It has been a pleasure to act as a reviewer for your mansucript titled ‘A Unique Bilateral Variation of the Extensor Carpi Radialis Longus’, which describes in detail an anatomic variation found through anatomic dissection. The case is well described, well illustrated as well as concise and adequately discussed. Although I believe it to be publishable in its present form, it would be greatly benefitted of a propsective Fig.1 showing a drawing or schematic diagram of your variation in comparison to the canonical patternd and ECRI, ECRA, ECRT. Also, some classical citations are lacking (e.g: Wood J: On some variations in human myology. Roy Soc Proc 15:229-244, 1867; Albright, J. A., & Linburg, R. M. (1978). Common variations of the radial wrist extensors. The Journal of Hand Surgery, 3(2), 134–138. doi:10.1016/s0363-5023(78)80060-4). A brief sentences explaining the conservation method would be also a good improvement.

Un saludo.

Round 2

Reviewer 1 Report

Comments and Suggestions for Authors

The most current article that deals with your topic is : 

Translational Research in Anatomy

Volume 35, June 2024, 100287 Examination of accessory extensor carpi radialis longus and brevis musculotendinous units for functional impact and tendon transfer suitability Jay J. Byrd a, Andrew C. White a, Travis L. McCumber a, Ethan L. Snow b https://doi.org/10.1016/j.tria.2024.100287 I strongly recommend its inclusion

Author Response

The most current article that deals with your topic is : 

Translational Research in Anatomy

Volume 35, June 2024, 100287 Examination of accessory extensor carpi radialis longus and brevis musculotendinous units for functional impact and tendon transfer suitability Jay J. Byrd a, Andrew C. White a, Travis L. McCumber a, Ethan L. Snow b https://doi.org/10.1016/j.tria.2024.100287 

I strongly recommend its inclusion

Response: Agreed, and thank you for the citation! A sentence and the citation have been included in the text (lines 58-60, in Introduction, highlighted in yellow); the citation has also been added to the references (now #13, highlighted in yellow). All subsequent reference numbers have also been adjusted.